# Examining the Relationship between Transportation Infrastructure, Urbanization Level and Rural-Urban Income Gap in China

Meseret Chanieabate [1], Hai He [1,*], Chuyue Guo [2], Betelhem Abrahamgeremew [3] and Yuanji Huang [1]

[1] Research Institute of Rural Revitalization, Hunan University of Science and Engineering, Yongzhou 425199, China
[2] School of Economics, Hunan Agricultural University, Changsha 410128, China
[3] School of Public Administration, Hunan Agricultural University, Changsha 410128, China
* Correspondence: hehai@huse.edu.cn; Tel.: +86-15773158953

**Abstract:** The development of transportation infrastructure plays a pivotal role in the regional economy from multiple dimensions. The aim of this paper is to examine the relationship between transportation infrastructure development and income inequality in urban and rural areas of China. The study utilizes panel data from 30 provinces, spanning the years 2010 to 2020, and employs the spatial Dubin model to measure and test the impact of transportation infrastructure on the urban-rural income gap. Furthermore, an intermediary effect test method is used to investigate the potential mediating effect of urbanization in this relationship. The results indicate that transportation infrastructure has a significantly negative direct, indirect, and total effect on the urban-rural income gap, with the indirect effect being greater than the direct effect. This suggests that transportation infrastructure can effectively reduce income disparities, with a noticeable spatial spillover effect. The level of urbanization plays a significant intermediary effect on the effect of transportation infrastructure on the urban-rural income gap, highlighting the role of transportation infrastructure in improving urbanization and narrowing income disparities. These findings underscore the importance of enhancing both the level of urbanization and cooperation between neighbouring regions in order to maximize the benefits of transportation infrastructure development for reducing income disparities and promoting regional balance in China.

**Keywords:** transportation infrastructure development; rural-urban income gap: urbanization; spatial inequality; spatial Dubin model; intermediary effect model

## 1. Introduction

Transportation infrastructure development in China is vital for economic growth, promoting social welfare, and achieving regional balance [1,2]. It provides physical connectivity through roads, railways, ports, and airports, facilitating trade, investment, and commerce. This enables access to markets, reduces transportation costs, and promotes competition, leading to economic growth [3,4]. Transportation infrastructure also plays a crucial role in enhancing access to education, healthcare, and other essential services, especially in rural areas that lack such amenities. Additionally, transportation infrastructure development helps bridge the gap between urban and rural areas, reducing income disparities and promoting regional equalization [5]. Investment in transportation infrastructure leads to increased efficiency, greater connectivity, and enhanced access to opportunities, contributing to sustainable development in both rural and urban areas [6].

The transportation infrastructure development in China has been characterized by significant investment and expansion over the past several decades [7]. In the 1980s, the Chinese government embarked on a series of reforms to modernize and improve the country's transportation infrastructure, which had been neglected during the Cultural

Revolution [8]. These reforms included decentralization of transportation planning and management, as well as the establishment of new institutions to oversee transportation infrastructure development at the provincial and local levels. Since then, China has made massive investments in transportation infrastructure development, particularly in large-scale projects such as the high-speed rail network and expressways [9]. This investment has helped to transform the country's transportation system, greatly increasing connectivity and reducing travel times between major urban centers.

However, there have been persistent disparities in transportation infrastructure investment and development between urban and rural areas. Urban areas have received a disproportionately larger share of investment in transportation infrastructure, leading to a significant urban-rural divide in terms of access to transportation services [10]. This has contributed to widening income disparities between urban and rural areas, as urban areas have benefited from greater access to opportunities and growth while rural areas have lagged behind [11]. Despite these challenges, recent years have witnessed increased investment in transportation infrastructure in rural areas, with a focus on improving connectivity and reducing the urban-rural gap. For example, the "New Rural Construction" strategy has been implemented to promote comprehensive rural development, including investments in transportation infrastructure (China releases action plan on rural construction, https://english.www.gov.cn/policies/latestreleases/202205/24/content_WS628c31bcc6d02e533532b372.html, accessed on 26 March 2023). These efforts aim to enhance the competitiveness of rural industries, increase access to markets, and promote regional balance [12]. Transportation infrastructure development in China has been marked by significant investment and expansion, but has also been characterized by persistent urban-rural disparities. Efforts to address these disparities through increased investment and optimized planning approaches represent a critical step toward promoting balanced economic development and reducing income disparities in China.

Transportation infrastructure development is a crucial factor affecting regional economic growth and development. However, little is known about the relationship between transportation infrastructure development and income inequality in urban and rural areas of China. This study investigate the mechanisms and effects of transportation infrastructure development in China towards narrowing the income disparities between urban and rural areas. The study seeks to answer the following research questions: What is the impact of transportation infrastructure on the urban-rural income gap in China? How does the level of urbanization mediate the relationship between transportation infrastructure and the urban-rural income gap? What are the pathways through which transportation infrastructure can be used to narrow the income disparities between urban and rural areas in China? This study utilizes a Spatial Dubin model and an intermediary effect test method to analyze the impact of transportation infrastructure on China's urban-rural income gap, and to investigate the mediating role of urbanization level. The scope of this study covers a time period from 2010 to 2020 and spans 30 provinces in China. The study uses panel data for all 30 provinces in China, allowing for a more comprehensive analysis of the impact of transportation infrastructure development on the urban-rural income gap. Analyze the mechanisms by which transportation infrastructure can be used to narrow income disparities between urban and rural areas, and provide recommendations for policymakers to optimize the impact of infrastructure investments on regional balance in China.

This study has significant implications for policy-makers, practitioners, and scholars interested in transportation infrastructure development and income disparities between China's urban and rural areas. Empirically, the research provides evidence for the effectiveness of transportation infrastructure in promoting regional balance and reducing income disparities. Theoretically, the findings enhance our understanding of the complex relationship between transportation infrastructure, urbanization, and income disparities. The study offers recommendations for optimizing the impact of transportation infrastructure investments on regional balance, including the need to strengthen cooperation between neighbouring regions and boost urbanization levels. These insights can inform

policymakers seeking to promote balanced economic development and reduce urban-rural disparities both in China and other countries with similar challenges. The existing literature on the rural-urban income gap in China tends to overlook the relationship between transportation infrastructure, urbanization, and income disparities. This is a significant gap in the literature, as it fails to consider the interdependent nature of these factors and the potential implications of their integration. Addressing this gap by examining the interplay between transportation infrastructure, urbanization, and income disparities from multiple perspectives could provide a more comprehensive understanding of the issue and inform more effective policies and strategies for reducing rural-urban income disparities in China.

The paper follows a rigorous analytical approach, with a Section 2 that explores existing studies on transportation infrastructure and income disparities. The Section 3 explains the sources, variables, and analysis methods employed. In the Section 4, the study finds that transportation infrastructure can be used to narrow income disparities by mediating the effect of urbanization level. Discussion focuses on policy recommendations for promoting regional balance and reducing income disparities. The paper concludes by summarizing the main findings and discussing limitations and future research directions. Overall, this study offers comprehensive empirical evidence and policy recommendations for policymakers, practitioners, and scholars interested in reducing income disparities in China.

## 2. Literature Review

### 2.1. Overview of Income Disparities between Urban and Rural Areas in China

The rural-urban income gap has been a longstanding challenge in China since the late 1970s economic reforms. While China has experienced significant economic growth and development over the past few decades, the income gap between urban and rural areas has widened [13]. Urban areas have been the primary beneficiaries of China's economic growth, resulting in higher incomes compared to rural areas due to factors such as land ownership, access to education, healthcare, and job opportunities [14]. Additionally, the hukou system divides the population into urban and rural categories, restricting access to public services based on one's hukou status [15]. Although the Chinese government has implemented policies to tackle the rural-urban income gap, it remains a significant challenge.

Numerous studies have investigated the factors that influence the income gap between rural and urban areas in China. One study discovered that as urbanization proceeded, both accumulated and flow income Gini coefficients declined, indicating a reduced rural-urban income gap [16]. These findings suggest that governments could implement policies that encourage urbanization to narrow the income gap between rural and urban areas. Conversely, another study revealed that the direct effect of migration on household income was negatively associated with local resource contributions, highlighting the importance of developing local resources to boost rural residents' income [17]. Entrepreneurial clusters, which are primarily composed of non-state-owned enterprises, are also capable of reducing local urban-rural income inequality by increasing rural residents' earnings. However, this clustering effect may be less pronounced in heavily urbanized regions or megacities [18]. A recent study identified that the Provincial Per Capita Net Urban-Rural Income Ratio (PPUR) exhibited high spatial agglomeration in Eastern China but low values in Central and Western China. The PPUR in the province was influenced by factors such as industrial structure, infrastructure, medical resources, and land-centered urbanization. The rural-urban income gap boosted the province's PPUR but hindered it in nearby provinces, indicating a need for policies to narrow the gap and reduce the PPUR [19].

The excessively large income gap poses a significant challenge to the quality of economic growth by influencing its foundation, operation, and outcome. Research has shown that investments in human and physical capital, improvements in transport infrastructure, industrial structure, and economic openness can play an active role in enhancing economic growth quality. On the other hand, government expenditure scale, financial development, and industrial structure deviation have a negative effect [20]. Moreover, regional disparities

and the rural-urban gap contribute substantially to China's high income inequality. Although reductions in rural poverty appear to be more effective in reducing both urban and rural poverty, the costs of achieving these reductions have not been fully considered [21]. Overall, the existing literature highlights the complexity of the rural-urban income gap in China and emphasises the need for evidence-based policies to reduce this gap, which requires attention from policymakers, researchers, and practitioners alike [22].

### 2.2. Transportation Infrastructure Development and Its Role in Reducing Rural Urban Income

Transport infrastructure plays a vital role in explaining the economic growth gap between regions in China [12]. Research indicates that areas with good transport facilities experience higher economic growth rates than those with poor infrastructure [23]. The positive effects of transport infrastructure are also evident during recession periods, where access to intermodal services and local and regional markets contribute to regional performance [24]. Furthermore, studies have shown that public infrastructure accessibility is positively associated with food security among rural households [25]. However, the impact of transport infrastructure expenditure varies greatly across countries, highlighting the need for assessing the specific context of each country when designing sustainable transportation plans [26]. Transport infrastructure has also been found to affect employment density in the service industry, with roads promoting employment more than railways and inland waterways [27]. Additionally, transportation infrastructure investment under the Belt and Road Initiative has generated varying impacts among different regions, highlighting the importance of assessing the effectiveness of transportation infrastructure investments based on context [6].

Investors, policymakers, and government agencies can estimate the potential outcomes of proposed transportation investment plans through modeling the complex interactions between transportation infrastructure, economic growth, and other factors, and develop optimal policies for transportation investment [28]. Furthermore, the coordinated development levels of economic, social, and environmental benefits in urban areas need improvement through the promotion of public transportation infrastructure [29]. Continuous investment in transportation infrastructure sectors is essential to achieve high levels of economic growth [30]. However, the lack of infrastructure maintenance eliminates the positive effects of investments over time, particularly in the medium term [31]. Transport infrastructure developments have a beneficial distributive effect, helping to promote inclusive growth in rural areas and reduce income inequality [12,32]. Nonetheless, transport poverty, caused by inadequate transport options, the hukou system, and jobs-housing imbalance, needs to be addressed to promote equitable access to transportation and support sustainable development [33]. In conclusion, the research highlights the crucial role of transport infrastructure in promoting economic growth and other aspects of sustainable development. Nonetheless, policymakers need to assess the specific context of each region or country when designing sustainable transportation infrastructure plans to maximize positive outcomes.

Empirical researchers have extensively studied the influence of transport infrastructure on the urban-rural income gap in China. Despite their effectiveness in reducing the gap, high-speed railways are not solely responsible for the formation of the three convergence clubs [34]. The impact of high-speed railways on narrowing the income disparity remains limited. Instead, national, provincial, and municipal roads play a significant role in reducing the urban-rural income gap in China [11]. These roads facilitate rural labor mobility, providing access to local and regional job markets for migrant workers. Moreover, road infrastructure is particularly crucial for boosting the income of rural residents in China's southwestern and middle regions. With road access, these residents can participate in economic activities and increase their earnings. Consequently, roads are an essential factor in reducing the urban-rural income gap in these regions.

Improving the provision of public services, specifically "soft" public services like education, medical care, and social security, is also an effective way to narrow the urban-

rural income gap in China [35]. Road investments, especially in rural areas, have a positive impact on overall economic growth and can lead to poverty reduction in both urban and rural areas [36]. Additionally, the agglomeration of producer services can play a vital role in narrowing China's urban-rural income gap [37]. Investment in rural infrastructure, particularly in agriculture and transportation, can alter land efficiency and the structure of land use and, as a result, indirectly impact farmers' income [38]. However, the effect of rural highways on the income gap among farmers across provinces follows a "U-shaped" curve, indicating a more significant initial impact that declines with further investment [39]. These effects are more pronounced for workers in non-state-owned enterprises, migrants seeking jobs via market method, and migrants working in high labor-intensive industries [33].

There are also potential negative consequences to consider when investing in transportation infrastructure. For example, the development of industrial agglomeration between cities may come at the expense of further increasing the urban-rural income gap [40]. Additionally, high-speed rail exacerbates health inequalities among high-income groups, highlighting the need for comprehensive policies that address access to healthcare services and social capital [41]. In conclusion, transport infrastructure has a significant role to play in narrowing the urban-rural income gap in China. However, policymakers need to carefully consider the potential benefits and drawbacks of different types of infrastructure investments and ensure that their policies are designed to be equitable and inclusive.

### 2.3. Theory on the Mechanisms and Effects of Transportation Infrastructure on Income Distribution

The oft-repeated adage "If you want to be rich, build roads first" (Huaxia (22 September 2019). Wanna be rich? Build roads first! Explore #HowChinaCan put every village on the right track http://www.xinhuanet.com/english/2019-09/22/c_138411531.htm, accessed on 22 February 2022) highlights the critical role of convenient transportation infrastructure as a prerequisite for economic development. In China, transportation infrastructure has been a key driver of economic growth [28,31]. However, this growth has also highlighted several structural issues, including unreasonable income distribution and excessive income gaps, that have come under increasing scrutiny.

One way transportation infrastructure affects income distribution is through direct effects [39,41]. For example, transportation infrastructure involves an array of sectors from initial investment and construction to later maintenance and repair needs. Infrastructure construction contributes to industrial development [42], which can provide many employment opportunities for rural surplus labor, directly improving the income of rural residents [43]. Additionally, transportation infrastructure can facilitate the transportation of goods and services, reducing transaction costs and promoting economic activity, thus playing a vital role in shaping income distribution in both urban and rural areas. Understanding such direct effects is crucial in ensuring that transportation infrastructure investment promotes sustainable and equitable economic development.

The impacts of transportation infrastructure development on income levels are not limited to direct effects but also extend to indirect effects, which are mainly expressed in urban and rural areas [44]. The development of transportation infrastructure in urban areas deepens the division of labor among various industrial departments, improves commuting efficiency for urban residents, and saves time for other productive activities [45]. Additionally, it reduces transportation costs and promotes industrial structure adjustment among regions. This includes developed industries in cities that can drive the development of vulnerable industries in rural areas, promoting common prosperity.

Conversely, the improvement of transportation infrastructure in rural areas can promote market integration between developed urban areas and backward rural areas, driving the growth of the rural economy [5]. By improving transportation, it becomes easier to transport agricultural and sideline products, expand potential markets, and reduce transaction costs and uncertainties [12]. Moreover, it enables urban technology and human capital to enter and exit rural areas, facilitating experts and technicians to provide technical

guidance for rural production and life. The smooth traffic infrastructure is also beneficial in improving agricultural productivity, rural construction levels, and increasing the income levels of rural residents [46]. With transportation infrastructure's construction in different regions, urban and rural income levels of the region and adjacent regions will change through the spatial spillover effect [47]. Therefore,

**Hypothesis 1:** *Traffic infrastructure development directly narrows the urban-rural income gap and that there is a spatial spillover effect.*

The urban-rural income gap in China can largely be attributed to the urban-rural dual structure. Addressing this issue under a program of common prosperity is crucial to solving issues of income inequality and promoting inclusive economic growth. Improving the level of urbanization is seen as an important solution for eliminating this dual structure. One way urbanization can help reduce the urban-rural income gap is by promoting labor mobility [48]. An increase in rural residents working in cities and towns leads to a reduction in the rural labor force engaged in agricultural production, which in turn increases the price of agricultural products and drives corresponding increases in farmers' income. Urbanization also increases demand for agricultural products, further raising prices and promoting farmers' income. Alongside market factors, government policies have focused increasingly on rural areas in recent years, with investment, tax, price, public services, and social security measures aimed at promoting sustainable development in rural areas [49].

Transportation infrastructure is key to income levels, with labor transfer, industrial upgrading, and factor flow all important indicators of urbanization [11]. Improved road traffic connections can accelerate the transfer of rural labor to cities and towns, while facilitating the accumulation of human capital for rural residents. Connectivity between urban and rural areas has become increasingly convenient, helping advanced production technologies and factors radiate to remote rural areas through transportation infrastructure [12]. Industrial upgrading can promote the development of secondary and tertiary industries in rural areas, providing employment for rural residents and driving capital to the countryside. Additionally, transportation infrastructure can accelerate the transfer of production factors and financial capital between urban and rural areas, attracting more factors, capital, and industries, further closing the urban-rural income gap (Figure 1). Thus,

**Hypothesis 2:** *Transportation infrastructure can improve the level of urbanization and reduce the urban-rural income gap by accelerating the pace of urbanization, improving labor mobility and access to markets, and attracting capital and industry to rural areas.*

Figure 1 illustrates the complex interplay between three critical factors: transport infrastructure, rural-urban income, and urbanization. The graphic representation provides valuable insights into the relationship between these variables and their influence on one another. By visualizing this interdependence, policymakers and researchers can better understand the various dynamics at play and formulate more effective strategies to promote sustainable development and economic growth in both urban and rural areas.

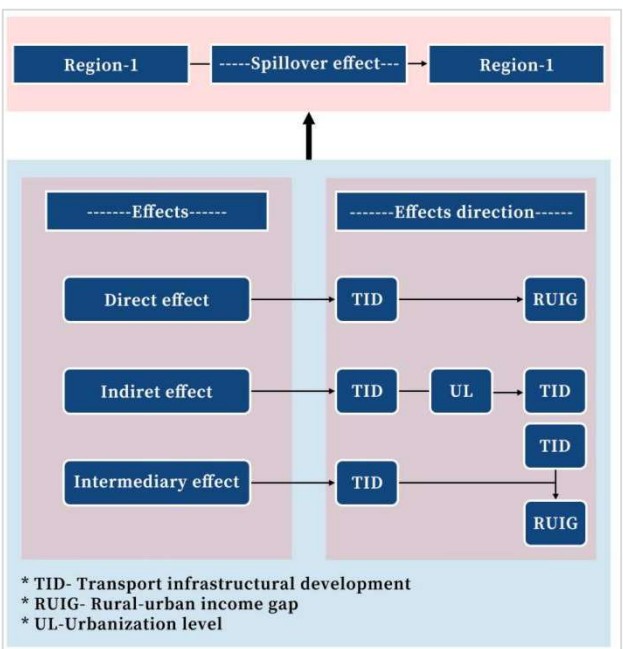

**Figure 1.** The theoretical framework.

### 3. Data and Methods

#### 3.1. Data Sources

This study utilizes panel data from 30 provinces, municipalities, and autonomous regions in China over a 20-year period from 2000 to 2020. These data sources were obtained from reputable organizations such as the National Bureau of Statistics, EPS database, the Ministry of Finance, and the State Administration of Taxation. These organizations are well-known for providing reliable and accurate data for academic research. It is important to note that the data from Tibet, Hong Kong, Macao, and Taiwan were not included in the analysis due to difficulties in obtaining these data. Therefore, the study's findings may not be representative of these regions, and the conclusions drawn from the analysis should not include these territories.

The use of panel data allows for the analysis of trends and changes over time in the different regions, providing a more comprehensive understanding of the phenomenon under investigation. By utilizing data from multiple sources, the study enhances its validity and reliability, as it is less likely to be biased by any one data source. Overall, the utilization of these high-quality data sources allows for a robust examination of the research question, providing insights that can inform policy-making and contribute to the academic literature.

#### 3.2. Variables

Explained variables: The urban-rural income gap, the urban-rural income ratio, and Theil index are commonly used in academic research to measure the disparity in income between urban and rural areas. This paper focuses on the urban-rural income ratio as the dependent variable. It shows that from 2000 to 2021, the urban-rural income gap followed an inverted U-shaped curve, with an initial upward trend, followed by a downturn (Figure 2). This finding is consistent with the Kuznets hypothesis on income distribution, which proposes that as an economy develops, income inequality initially increases before eventually declining. Thus, this study is interested in examining how various factors contribute to the changes in the urban-rural income ratio over time, using the urban-rural income gap as a starting point. By focusing on the urban-rural income ratio and considering the shifts in the urban-rural income gap, this study aims to provide insights into the dynamics of urban-rural economic development and the factors influencing these trends.

**Core explanatory variable:** The core explanatory variable in this research study is traffic infrastructure, specifically road infrastructure. Roads play a critical role in connecting rural and urban areas and are vital to overall economic development. They facilitate the movement of goods and people, which is essential to the production and sale of goods. In this study, highways were selected as the most representative type of traffic infrastructure for analysis. The ratio of total highway mileage to land area in each province, municipality, or autonomous region is used as the measure of road infrastructure. This ratio is a useful indicator of the extent of highway development in a given region, with higher ratios indicating more extensive infrastructure development. By including road infrastructure as a core explanatory variable, this study aims to examine how improvements in the quality and quantity of road infrastructure impact the levels of urban-rural income disparity in China over time. This information can provide valuable insights to policymakers and aid in the formulation of policies aimed at reducing such disparities.

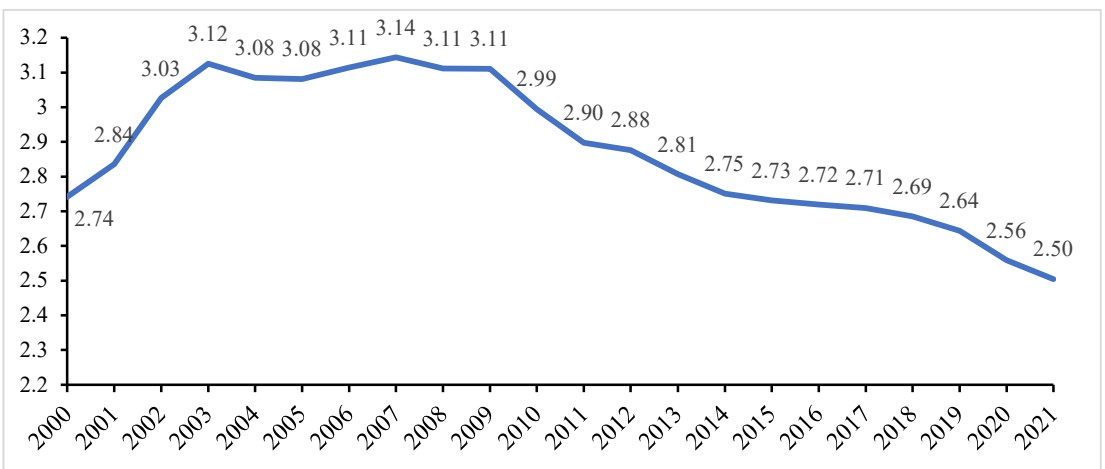

**Figure 2.** The annual average urban rural income gap change in China from 2000 to 2021 (Source: National Bureau of Statistics).

Intermediate variable: The intermediate variable used in this study is the level of urbanization, represented by the urbanization rate (UR). The urbanization rate is a commonly used indicator for measuring the level of urbanization by many scholars. It is calculated as the ratio of urban population to the year-end resident population. A higher urbanization rate indicates a greater proportion of a region's population living in urban areas. This variable is an essential factor in assessing and understanding the overall development of urban and rural areas. An increasing urbanization rate implies changes in economic activities, employment patterns, and social structures, which could impact income distribution and inequality between rural and urban regions. Therefore, this study uses the urbanization rate as an intermediate variable to analyze its impact on the relationship between road infrastructure and the urban-rural income ratio. By taking into account the level of urbanization, this study aims to provide a more comprehensive understanding of how road infrastructure and urbanization drive the dynamics of urban and rural economic development and their influence on the urban-rural income gap.

Control variables: This study includes several control variables to account for potential confounding factors that may impact the relationship between road infrastructure, urbanization, and the urban-rural income ratio. The first variable is the proportion of government expenditure on agriculture (AC), measured as the ratio of government expenditure on agriculture, forestry, water, and general government budget expenditure. This variable is important as it could potentially affect the overall economic growth and development of rural areas. The second variable is foreign trade openness (FTO), which is measured by the ratio of total imports and exports to GDP. A higher FTO ratio suggests a higher degree of economic integration with other countries, potentially resulting in variations in regional

development and income distribution. The third variable is agricultural development level (DR), which is expressed by the grain disaster rate, a measure of the area affected by natural disasters relative to the area designated for grain production. This variable is included as it could potentially affect agricultural productivity and cause disparities in rural incomes. The fourth variable is rural infrastructure level (HEAL), which is represented by the number of village and town clinics. Greater investment in healthcare infrastructure in rural areas can improve the health and well-being of rural residents, potentially contributing to economic growth and reduced income inequality. Finally, gross domestic product (GDP) growth rate is included as a control variable, representing the overall economic development of the region. By including these control variables, this study aims to account for potential confounding factors that might otherwise affect the relationship between road infrastructure, urbanization, and the urban-rural income ratio. Descriptive statistics of each variable are presented in the study for thorough analysis.

Table 1 provides information on the variables in the study and their descriptive statistics. The selection of variables in this study is based on existing research on urban-rural income disparities, with previous studies indicating that road infrastructure, urbanization, government expenditure on agriculture, foreign trade openness, agricultural development, rural infrastructure, and economic growth are key factors in determining income gaps in China. Road infrastructure is particularly important in connecting rural and urban areas, while urbanization rates could signal changes in economic activity, employment patterns, and social structures that impact income inequality. The inclusion of control variables, such as government expenditure on agriculture, foreign trade openness, agricultural development level, rural infrastructure level, and GDP growth rate, further enhances the study's reliability and comprehensiveness in analysing the dynamics of urban and rural economic development and their effect on the urban-rural income gap in China.

**Table 1.** Variables and statistical descriptions.

| Variable | Definition | N | Mean | SD | Min | Max |
|----------|-----------|---|------|-----|-----|-----|
| Gap | The disposable income of urban residents divided by that of rural residents | 330 | 2.69 | 0.45 | 1.84 | 4.07 |
| Road | The ratio of total highway mileage to land area | 330 | 0.90 | 0.51 | 0.05 | 2.21 |
| UR | The ratio of urban population to year-end resident population | 330 | 0.56 | 0.13 | 0.33 | 0.94 |
| AC | The ratio of government expenditure on agriculture, forestry, water, and general government budget expenditure | 330 | 0.12 | 0.03 | 0.04 | 0.20 |
| FTO | Total imports and exports divided by GDP | 330 | 0.28 | 0.31 | 0.01 | 1.61 |
| DR | Grain disaster rate (The ratio of grain disaster area to grain sown area) | 330 | 0.17 | 0.11 | 0 | 0.61 |
| HEAL | The number of village and town clinics. | 330 | 21,092.18 | 17,022.11 | 1162 | 66,277 |
| GDP | The growth rate of regional GDP | 330 | 0.10 | 0.07 | −0.25 | 0.30 |

### 3.3. Spatial Dubin Model

The Spatial Dubin Model (SDM) is a type of spatial point process model that was first introduced by Dubin (1978) to study the distribution of crime incidents in Los Angeles. It has since been applied to a wide range of other areas, including ecology, epidemiology, and transportation. The spatial Dubin model assumes that the underlying spatial point process of interest is a Poisson point process with an intensity function that depends on both the distance between points and the covariates associated with the points [50]. The model also assumes that the distances between points follow a bivariate normal distribution, and that the covariance structure of the distance distribution can be described by a correlation parameter. Let Y be a spatial point process with points located in a region D. The spatial Dubin model assumes that the log of the intensity function of Y can be represented as a linear combination of covariates X and a distance function d(Y), i.e.,

$$\log(\lambda(Y)) = X\beta - \phi d(Y)$$

where $\lambda(Y)$ is the intensity function of the point process, $\beta$ is a vector of regression coefficients, $\phi$ is a parameter that controls the strength of the distance effect, and $d(Y)$ is the minimum distance between any pair of points in the set Y.

There are several methods for estimating the parameters of the spatial Dubin model, including maximum likelihood estimation (MLE), Bayesian inference [51], and Markov chain Monte Carlo (MCMC) methods [52]. MLE involves maximizing the likelihood of the observed data under the spatial Dubin model, while Bayesian inference involves determining the posterior distribution of the model parameters given the data and prior information. MCMC methods generate samples from the posterior distribution using iterative simulation, and can be used to estimate both the parameters and their uncertainties. In summary, the spatial Dubin model is a flexible and useful tool for analyzing spatial point patterns that incorporates both covariate effects and distance effects. Its assumptions, mathematical formulation, and estimation methods make it applicable to a wide range of fields and research questions.

According to existing research, there is a strong spatial correlation between transport infrastructure and the urban and rural income gap in China. To accurately model this spatial correlation, we turned to the spatial Dubin model (SDM), which includes both endogenous and exogenous interaction models. Unlike other spatial econometric models, the SDM can account for spatial correlation when variables are missing, leading to more precise regression results. For this reason, we used the SDM to test our regression.

$$\mathrm{Gap}_{ij} = \beta_0 + \rho\sum\nolimits_j w_{ij}\mathrm{Gap}_{ij} + \beta_1 \mathrm{Road}_{it} + \lambda_1 \sum\nolimits_j w_{ij}\mathrm{Road}_{it} + \beta_2 X_{it} + \lambda_2 \sum\nolimits_j w_{ij}X_t + \mu_i + \sigma_t + \varepsilon_{it} \quad (1)$$

Our regression model, as shown in Equation (1), examines the relationship between the urban-rural income gap (Gap) and road infrastructure (Road) as well as other control variables (X). We employed a spatial weight matrix ($W_{ij}$) to capture the spatial dependencies among our observations. Additionally, we included individual effects ($\mu_i$), time effects ($\sigma_t$), and an error term ($\varepsilon_{it}$) to control for unobserved heterogeneity and measurement errors.

Our study covers 30 provincial administrative regions in China (excluding Tibet, Hong Kong, Macao, and Taiwan) from 2010 to 2019. By leveraging the SDM, we aim to develop a better understanding of the impact of road infrastructure on the urban-rural income gap. Through our analysis, we hope to provide insights that can inform policy decisions aimed at reducing income disparities in urban and rural areas.

### 3.4. Intermediary Effect Model

The intermediary effect model is a statistical technique used to explore how an intermediate variable mediates the relationship between an exposure variable and an outcome variable [53]. In other words, it examines how much of the effect of the exposure variable on the outcome variable is explained by changes in the intermediate variable. The intermediary effect model assumes that the exposure variable affects the intermediate variable, which in turn affects the outcome variable. It also assumes that there are no unmeasured confounders that can influence both the exposure and outcome variables.

Suppose we are interested in studying the relationship between an exposure variable X, an intermediate variable M, and an outcome variable Y. The intermediary effect model can be expressed using the following regression equations:

$$M = \alpha_0 + \alpha_1 X + \varepsilon_1$$
$$Y = \beta_0 + \beta_1 X + \beta_2 M + \varepsilon_2$$

where $\alpha_0$, $\alpha_1$, $\beta_0$, $\beta_1$, and $\beta_2$ are coefficients to be estimated, and $\varepsilon_1$ and $\varepsilon_2$ are error terms. The coefficient $\beta_2$ represents the direct effect of the intermediate variable M on the outcome variable Y, while the product of $\alpha1$ and $\beta_2$ represents the indirect effect of the exposure variable X on the outcome variable Y through the mediator M. The total effect of X on Y is the sum of the direct and indirect effects.

There are several methods for estimating the parameters of the intermediary effect model, including least squares regression and structural equation modeling (SEM) [54]. Least squares regression involves regressing both the intermediate and outcome variables on the exposure variable, and then calculating the indirect effect as the product of the corresponding coefficients. SEM is a more flexible approach that allows researchers to specify more complex models and test for additional hypotheses. In summary, the intermediary effect model is a useful tool for investigating how an intermediate variable mediates the relationship between an exposure variable and an outcome variable. Its assumptions, mathematical formulation, and estimation methods make it applicable to a variety of research questions in fields such as epidemiology, psychology, and social sciences.

In this study, we employed intermediary effect model to analyze how an intermediate variable mediates the relationship between an exposure variable and an outcome variable. In the context of transportation infrastructure, urbanization level can act as an intermediary variable that affects income disparities between urban and rural areas in China. The equations of intermediary effect model in this study is;

$$UR_{it} = \alpha_0 + \rho_m \sum_j w_{ij} U_{it} + \alpha_1 Road_{it} + \Phi_1 \sum_j j w_{ij} Road_{it} + \alpha_2 X_{it} + \Phi_2 \sum_j w_{ij} X_{it} + \mu_i + \sigma_t + \varepsilon_{it} \quad (2)$$

where, $UR_{it}$ is the intermediary variable: urbanization level. The intermediary effect test steps are as follows. Step 1, test Equation (1), if the regression coefficient $\beta_1$ Significant, indicating that the improvement of transport infrastructure can directly narrow the urban-rural income gap; Otherwise, stop the inspection. Step 2: Test Equations (2) and (3), if the regression coefficient $\alpha_1$ and $\gamma_2$ are significant, indicating that there is a mediating effect; If the regression coefficient $\alpha 1$ and $\gamma$ If there is an insignificant value in 2, it needs to be further tested with Bootstrap method.

$$Gap_{it} = \gamma_0 + \rho_N \sum_j w_{ij} Gap_{it} + \gamma_1 Road_{it} + \delta_1 \sum_j w_{ij} Road_{it} + \gamma_2 UR_{it} + \delta_2 \sum_j w_{ij} UR_{it}$$
$$+ \gamma_2 X_{it} + \delta_3 \sum_j w_{ij} X_{it} + \mu_i + \sigma_t + \varepsilon_{it} \quad (3)$$

where, $Gap_{it}$ is rural-urban income gap,

In addition, if the regression coefficient in Equation (3) $\gamma_1$ Not significant $\gamma_2$. Significant, indicating that the level of urbanization has a complete intermediary effect; If the regression coefficient $\gamma_1$ and $\gamma_2$ are significant, indicating that the level of urbanization has a partial intermediary effect.

## 4. Results and Discussions

### 4.1. Spatial Correlation Test

To employ the spatial Dubin model for regression, it is necessary to test whether the urban-rural income gap exhibits spatial correlation by using the Moran index. The calculation formula for the Moran index is presented below as formula (4):

$$Moran's\ I = \frac{\sum_{i=1}^n \sum_{j=1}^n w_{ij} (Y_i - \overline{Y})(Y_j - \overline{Y})}{S^2 \sum_{i-1}^n \sum_{j=1}^n w_{ij}} \quad (4)$$

In the above formula, S2 denotes the sample variance, which represents the sample mean while $Y_i$ and $Y_j$ represents the observations of the ith and jth regions, respectively. Additionally, $w_{ij}$ is the spatial weight matrix. Table 2 presents the global Moran's I index and the corresponding statistical test results. It indicates the Moran's I index of the urban-rural income gap between 2010 and 2020. It is clear from the table that the urban-rural income gap exhibits a significant level of spatial correlation, as indicated by the Moran's I index, suggesting that neighboring regions with similar income levels are likely to cluster together. This information is essential for informing the appropriate modeling techniques and ensuring the validity of the spatial Dubin model in analyzing the relationship between the urban-rural income gap and various explanatory variables.

The analysis of the Moran's I index for urban-rural income gap between 2010 and 2020 suggests a significant level of spatial correlation, with an overall value that is significantly positive at a 1% significance level. This indicates that changes in the urban-rural income gap within each region are positively correlated on a global scale, with neighboring regions exhibiting similar urban-rural income levels. In Table 2, the range of Moran's I indices is from 0.258 to 0.402, which suggests a moderate to strong positive spatial autocorrelation of the urban-rural income gap across provinces in China.

Moreover, the overall Moran's I index for urban-rural income gap reveals a fluctuating downward trend between 2010 and 2020, suggesting that the spatial dependence of changes in the urban-rural income gap across different regions is gradually weakening over time. This observation highlights a potential shift towards greater income equality between urban and rural areas in China. However, it is important to note that despite a reduction in the strength of spatial dependence, there still remains a significant level of spatial correlation. Hence, further research is needed to investigate the underlying factors driving the observed changes in the urban-rural income gap and identify measures to promote equitable development across regions in China.

**Table 2.** The global Moran's I index of urban-rural income gap from 2010 to 2020.

| Time | Moran's I Index | *p*-Value | Average Values of the Moran Index |
|------|-----------------|-----------|-----------------------------------|
| 2010 | 0.402 | 0.000 | 0.306 |
| 2011 | 0.389 | 0.000 | 0.306 |
| 2012 | 0.386 | 0.000 | 0.306 |
| 2013 | 0.307 | 0.000 | 0.306 |
| 2014 | 0.295 | 0.000 | 0.306 |
| 2015 | 0.272 | 0.000 | 0.306 |
| 2016 | 0.258 | 0.000 | 0.306 |
| 2017 | 0.261 | 0.000 | 0.306 |
| 2018 | 0.258 | 0.000 | 0.306 |
| 2019 | 0.265 | 0.000 | 0.306 |
| 2020 | 0.275 | 0.000 | 0.306 |
| 2010 | 0.402 | 0.000 | 0.306 |
| 2011 | 0.389 | 0.000 | 0.306 |
| 2012 | 0.386 | 0.000 | 0.306 |
| 2013 | 0.307 | 0.000 | 0.306 |
| 2014 | 0.295 | 0.000 | 0.306 |
| 2015 | 0.272 | 0.000 | 0.306 |
| 2016 | 0.258 | 0.000 | 0.306 |
| 2017 | 0.261 | 0.000 | 0.306 |
| 2018 | 0.258 | 0.000 | 0.306 |
| 2019 | 0.265 | 0.000 | 0.306 |
| 2020 | 0.275 | 0.000 | 0.306 |

*4.2. Local Correlation Test*

While global correlation provides insights into the correlation of a given space as a whole, it may overlook spatial heterogeneity in local areas. To overcome this limitation, local spatial autocorrelation measures can be employed to evaluate different spatial aggregation patterns that potentially exist across disparate regions. The precise calculation formula for local spatial autocorrelation is presented as Equation (5):

$$\text{Moran's } I_i = \frac{(X_i - \overline{X}_i) \sum_{j=1}^{n} W_{ij}(X_j - \overline{X}_j)}{\frac{1}{n} \sum_{i=1}^{n} (X_i - \overline{X}_i)^2} \tag{5}$$

Specifically, $W_{ij}$ denotes the spatial weight matrix; $X_i$ and $X_j$ represent the attribute values of a given region I and its corresponding neighboring region j, respectively. Moreover, $X_i$ and $X_j$ are indicators of expected average values for these attributes. Additionally, N represents the total number of provinces or regions included in the study.

Two Moran scatterplots, Figures 3 and 4, were generated based on Equation (5) to analyze the spatial patterns of the urban-rural income gap in 2010, 2014, 2017, and 2020. As shown in the figures, the Moran scatter map is divided into four quadrants. Quadrants I and III indicate that the observations in a given region are similar to those of its surrounding areas, while Quadrants II and IV suggest that they differ.

Specifically, quadrant I corresponds to "high high" clustering, meaning regions with relatively high urban-rural income levels cluster together. Quadrant II is indicative of "low high" clustering, where regions with low urban-rural income gaps form clusters next to regions with high gaps. Quadrant III corresponds to "low low" clustering, suggesting that regions with low urban-rural income gaps are clustered together. Finally, quadrant IV corresponds to "high low" clustering, indicating that regions with high urban-rural income gaps cluster together.

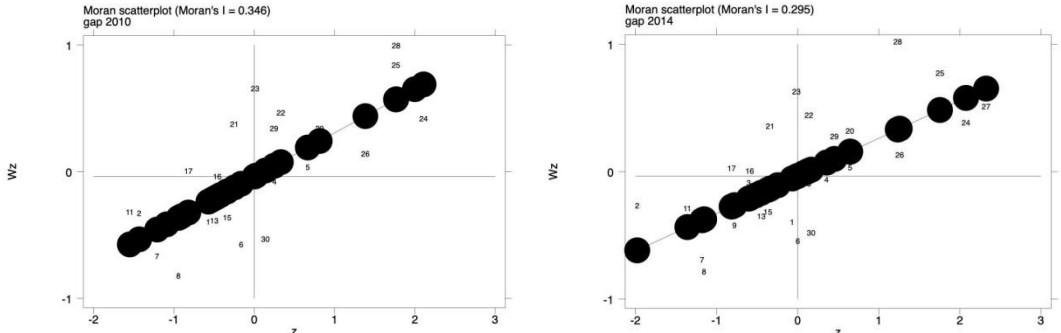

**Figure 3.** Moran Scatter of Urban Rural Income Gap in 2010 and 2014. Note: Beijing—1, Tianjin—2, Hebei—3, Shanxi—4, Inner Mongolia—5, Liaoning—6, Jilin—7, Heilongjiang—8, Shanghai—9, Jiangsu—10, Zhejiang—11, Anhui—12, Fujian—13, Jiangxi—14, Shandong—15, Henan—16, Hubei—17, Hunan—18, Guangdong—19, Guangxi—20, Hainan—21, Chongqing—22, Sichuan—23, Guizhou—24, Yunnan—25, Shaanxi—26, Gansu—27, Qinghai—28, Ningxia—29, Xinjiang—30.

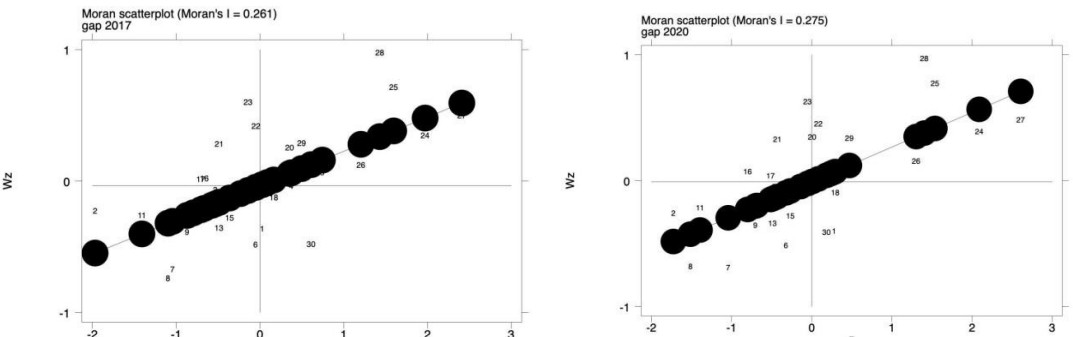

**Figure 4.** Moran Scatter of Urban Rural Income Gap in 2017 and 2020. Note: 1—Beijing, 2—Tianjin, 3—Hebei, 4—Shanxi, 5—Inner Mongolia, 6—Liaoning, 7—Jilin, 8—Heilongjiang, 9—Shanghai, 10—Jiangsu, 11—Zhejiang, 12—Anhui, 13—Fujian, 14—Jiangxi, 15— Shandong, 16—Henan, 17—Hubei, 18—Hunan, 19—Guangdong, 20—Guangxi, 21—Hainan, 22—Chongqing, 23—Sichuan, 24—Guizhou, 25—Yunnan, 26—Shaanxi, 27—Gansu, 28—Qinghai, 29—Ningxia, 30—Xinjiang.

As observed in both Figures 3 and 4, most of the data points fall in quadrants I and III, implying that regions with either large or small urban-rural income gaps tend to form clusters in space. This suggests that urban-rural income gap clustering is more common than regions that exhibit a mix of urban-rural income gap levels. These observations have important implications for policymakers seeking to promote regional equality and development in China.

### 4.3. Spatial Spillover Effect Test

Before proceeding to regression, Wald and LR tests were conducted to ascertain whether the original hypothesis could be simplified into either a spatial panel error model or spatial panel lag model. The tests conducted showed that the original hypothesis was rejected at a 1% significance level, indicating that neither of the models are suitable for analysis. Furthermore, a Hausman test was carried out and at a significant level of 1%, it rejected the original hypothesis, suggesting that the random effect could not be considered. Therefore, to address this issue, this paper selected the spatial Dubin fixed effect model as the most appropriate approach for the analysis. This model is likely to provide more accurate results in terms of evaluating the potential spatial spillover effects involved in the urban-rural income gap across different regions in China.

Based on the results presented in Table 3, it is evident that the direct and indirect effects of transportation infrastructure on the urban-rural income gap are negative and significant. Furthermore, the indirect effect, which represents the spatial spillover effect, is significantly greater than the direct effect. The indirect effect coefficient has a value of $-0.9065$ and is statistically significant at a 1% level, unlike the comparatively smaller direct effect of $-0.099$. These findings suggest that transportation infrastructure has a substantial spatial spillover effect in reducing the urban-rural income gap. These results corroborate the findings of a great deal of the previous work in spatial spillover effects of transport infrastructure [47]. Specifically, infrastructure projects implemented in a given region can have a ripple effect on surrounding areas, contributing to a decrease in the income gap between urban and rural residents. These results are consistent with hypothesis H1, demonstrating the potential for transportation infrastructure development as an effective strategy for promoting regional equity and bridging income gaps across China.

**Table 3.** Regression Results of Spatial Dubin Model.

|  | Model 1 | | |
|---|---|---|---|
|  | lnGAP | | |
|  | **Main Effect** | **Direct Effect** | **Indirect Effect** |
| lnRoad | −0.0847 ** | −0.0990 *** | −0.9065 *** |
|  | (−2.184) | (−2.577) | (−4.512) |
| lnAC | 0.0584 *** | 0.0567 *** | 0.0210 * |
|  | (2.859) | (3.262) | (1.851) |
| lnFTO | −0.0601 *** | −0.0601 *** | −0.0229 * |
|  | (−7.086) | (−6.629) | (−1.880) |
| lnDR | 0.0028 | 0.0031 | 0.0011 |
|  | (1.001) | (1.002) | (0.806) |
| lnHEAL | −0.1746 *** | −0.1845 *** | −0.0695 |
|  | (−3.816) | (−3.436) | (−1.608) |
| lnGDP | 0.0224 | 0.0235 | 0.0105 |
|  | (0.524) | (0.582) | (0.528) |
| N |  | 30 | |
| Obs |  | 330 | |
| Fixed by province |  | yes | |
| Fixed year |  | yes | |

The symbols ***, **, and * represent statistical significance levels of 1%, 5%, and 10%, respectively.

### 4.4. Robustness Test

To enhance the credibility of the regression results derived from the spatial Dubin model, this study sought to assess their robustness using three different spatial weight matrices: w1 geographical adjacency matrix, $w_2$ inverse distance matrix, and w3 reciprocal square sum of geographical distance matrix. Table 4 provides an overview of the

regression outcomes obtained from the spatial Dubin model based on these different spatial weight matrices.

Table 4 reveals that all three spatial weight matrices produced positive spatial autoregressive coefficients (Spa rho) for both urban and rural income variables, which were significant at a statistical level of 1%. This indicates a significant positive spatial relationship between provinces in terms of their urban-rural income gaps, as well as a clear spatial spillover effect. Furthermore, the spatial lag term for transport infrastructure was found to be negative, suggesting a negative spatial spillover effect. Thus, the development of transport infrastructure in neighboring provinces can restrain the expansion of the urban-rural income gap in a given province, thereby playing a role in narrowing the gap. The results of the regression analysis, using various spatial weight matrices, reveal significant negative coefficients for transport infrastructure, with a statistical significance level of 1%. These findings suggest that investments in transport infrastructure can play a crucial role in reducing the income gap between urban and rural areas. This finding confirms the reliability and robustness of the spatial Dubin model applied in the study. This study supports evidence from previous observations [11,34].

**Table 4.** Regression results of different spatial weighting matrices.

| | Model 2 | | |
| --- | --- | --- | --- |
| | lnGAP | | |
| | **W1 (Geographic Adjacency Matrix)** | **W2 (Geographic Inverse Distance Matrix)** | **W3 (Matrix of Reciprocal Square Sum of Distance)** |
| lnROAD | −0.0847 ** | −0.0414 *** | −0.0313 *** |
| | (−2.184) | (−3.622) | (−2.915) |
| Controls | | YES | |
| rho | 0.6665 *** | 0.5989 *** | 0.4636 *** |
| | (9.543) | (6.319) | (7.242) |
| sigma2_e | 0.0082 *** | 0.0117 *** | 0.0110 *** |
| | (12.660) | (12.621) | (12.572) |
| N | 30 | 30 | 30 |
| Obs | 330 | 330 | 330 |

The symbols ***, **, and * represent statistical significance levels of 1%, 5%, and 10%, respectively.

### 4.5. Intermediary Effect of Urbanization

To obtain a clearer understanding of the direct and indirect effects of the spatial panel Dubin model's parameter estimates, further decomposition is required using the partial differential method. In Model 3, the results indicate that transportation infrastructure has a significant indirect effect on urbanization levels, implying that infrastructure development can stimulate urbanization in surrounding areas through a radiating effect. These results match those observed in earlier studies [55]. The study employed the stepwise regression method to examine the intermediary effect of the urbanization level based on Models 2, 3, and 4.

The intermediary effect test based on the geographical adjacency matrix ($w_1$) involved three key steps. In the first step, the study used a formula to examine the impact of transportation infrastructure on the urban-rural income gap, obtaining the first column of Model 2 in Table 4. The results demonstrated statistical significance at a level of 1%, indicating that transportation infrastructure can effectively reduce the urban-rural income gap, thereby supporting hypothesis H1. The second step involved utilizing equation 2 to evaluate the influence of transportation infrastructure on the urbanization level, resulting in the first column of Model 3 in Table 5. The findings revealed a statistically significant relationship at a level of 10%. This suggests that transportation infrastructure development can promote urbanization levels and supports the original hypothesis H2. For the final step, the study used equation 3 to examine the combined impact of transportation infrastructure and urbanization levels on the urban-rural income gap. The results demonstrated

statistical significance at a level of 1%, with regression coefficients showing a negative relationship. This indicates that urbanization levels play an intermediary role in this relationship, with the proportion of intermediary effects estimated to be α one γ 2/β 1 = 5.3%. Therefore, hypothesis H2 was verified: transportation infrastructure can improve the level of urbanization.

Transportation infrastructure can also contribute to narrowing the urban-rural income gap by accelerating the pace of urbanization, consistent with research conducted by Mishra and Agarwal (2019) [56]. Financial support for agriculture, as a control variable, has a positive impact on the urban-rural income gap. However, it may face several challenges in the process, such as the allocation of multiple projects, decentralized distribution of funds, and limited management capacities, which could lead to insufficient funding for the agricultural sector. In contrast, foreign trade appears to have a significant impact on reducing the income gap between urban and rural areas. Firstly, foreign trade creates employment opportunities for rural residents, thereby increasing their wage income. Secondly, it provides farmers with a larger market to sell their agricultural products, increasing their productive income. Thus, foreign trade can help boost the income of rural residents and narrow the urban-rural income gap.

**Table 5.** Test results of intermediary effect mechanism.

| | Model 3 | | | Model 4 | | |
|---|---|---|---|---|---|---|
| | lnUR | | | lnGAP | | |
| | **Main Effect** | **Direct Effect** | **Indirect Effect** | **Main Effect** | **Direct Effect** | **Indirect Effect** |
| lnRoad | 0.0651 * | 0.0470 | 0.9406 *** | −0.2591 *** | −0.2613 *** | −0.1203 * |
| | (1.649) | (1.081) | (8.521) | (−4.944) | (−4.911) | (−1.773) |
| lnUR | - | | | −0.0691 * | −0.0833 *** | −0.6093 *** |
| | | | | (−1.841) | (−2.670) | (−3.637) |
| lnAC | 0.0272 | 0.0253 | −0.0053 | 0.0649 *** | 0.0669 *** | 0.0298 * |
| | (1.323) | (1.438) | (−1.008) | (3.287) | (3.174) | (1.741) |
| lnFTO | 0.0212 ** | 0.0218 ** | −0.0043 | −0.0545 *** | −0.0543 *** | −0.0246 * |
| | (2.495) | (2.398) | (−1.241) | (−6.603) | (−6.142) | (−1.864) |
| lnDR | −0.0014 | −0.0011 | 0.0002 | 0.0025 | 0.0019 | 0.0010 |
| | (−0.484) | (−0.362) | (0.334) | (0.901) | (0.598) | (0.587) |
| lnHEAL | 0.1852 *** | 0.1774 *** | −0.0356 | −0.1270 *** | −0.1245 *** | −0.0553 |
| | (4.011) | (3.272) | (−1.423) | (−2.813) | (−2.872) | (−1.599) |
| lnGDP | −0.1234 *** | −0.1230 *** | 0.0242 | −0.0075 | −0.0071 | −0.0030 |
| | (−2.859) | (−3.029) | (1.395) | (−0.180) | (−0.179) | (−0.157) |
| N | | | 30 | | | |
| Obs | | | 330 | | | |
| Fixed by province | | | yes | | | |
| Fixed year | | | yes | | | |

The symbols ***, **, and * represent statistical significance levels of 1%, 5%, and 10%, respectively.

The effect of the crop disaster rate on the urban-rural income gap was found to be insignificant. This could be because high rates of crop disasters reduce farmers' enthusiasm to grow crops, resulting in a reduction in their income. Moreover, the coefficient of rural infrastructure development was found to be significantly negative, indicating that it can play an essential role in providing improved production and living conditions for farmers, thereby increasing their income and contributing towards narrowing the urban-rural income gap. Lastly, the study indicated that GDP has an insignificant impact on the urban-rural income gap, suggesting that GDP growth is more effective in promoting overall income growth among urban and rural residents than in reducing poverty.

## 5. Conclusions and Policy Implication

The objective of economic development is to achieve common prosperity for all, which involves improving the national income level and reducing the income gap between urban

and rural residents. This study provides an in-depth analysis of the current situation of the urban-rural income gap in China. Firstly, the study calculates the urban-rural income ratio, which reveals an inverted U-shaped curve, with the income gap peaking in 2007 and gradually decreasing thereafter. This indicates that recent policies on urbanization development and rural governance have begun to show positive results. Secondly, the study uses the Moran index to conduct a spatial correlation test, demonstrating significant spatial correlation in the urban-rural income gap levels. Regions with large or small income gaps tend to cluster together. Thirdly, based on different spatial weight matrices, robustness and spatial spillover tests were conducted, demonstrating that transport infrastructure is not only vital in narrowing the urban-rural income gap within a province but also promotes the reduction of the income gap in neighbouring provinces through spatial spillover effects. The robustness results further support this conclusion. Finally, the study examines the intermediary effect based on the level of urbanization. The results indicate that urbanization has a beneficial role in the impact of transport infrastructure on the urban-rural income gap, playing a part as an intermediary in the relationship.

These findings have important implications for policymakers aiming to promote the common prosperity of all people by narrowing the urban-rural income gap in China. It suggests that developing transport infrastructure and supporting rural governance can facilitate the narrowing of the income gap and contribute to the achievement of the goal of common prosperity. Based on the study's conclusions, several policy implications can be drawn. Firstly, policymakers must prioritize the construction of transportation infrastructure, specifically highway infrastructure development. It is essential to decrease the differences in transportation infrastructure construction between urban and rural areas and focus on constructing village and township roads that are closely linked to farmers, thereby enhancing rural travel conditions. Secondly, it is essential to invest in and construct highway infrastructure in areas where the urban-rural income gap is significantly high to increase the marginal income generated from transportation infrastructure construction. This will maximize the role of transportation infrastructure in narrowing the urban-rural income gap. Lastly, the study recommends strengthening the construction of other public infrastructure in rural areas and refining the function of urbanization. Policymakers must strive to break the urban-rural dual structure, promote equal distribution of economic growth benefits among all citizens, and establish a strong foundation for the achievement of common prosperity. Overall, these policy implications highlight the critical role of infrastructure development in reducing the urban-rural income gap and achieving common prosperity for all people in China.

The study has several limitations that could be addressed in future research. Firstly, the analysis only focuses on the impact of transport infrastructure on the urban-rural income gap and does not consider other factors that could also play a role, such as education, health care, or social welfare policies. Future studies could adopt a more comprehensive approach and explore the impact of multiple factors on the urban-rural income gap. Secondly, the study analyzes the spatial correlation and spillover effects of transport infrastructure on the urban-rural income gap, but it does not delve into the mechanisms through which these effects occur. Future research could investigate the mechanisms and pathways that link transport infrastructure and the urban-rural income gap to provide policymakers with more practical guidance. Thirdly, the study employs panel data from 2010 to 2020, which may not fully capture the impact of recent policies addressing the urban-rural income gap, such as the targeted poverty alleviation campaign. Using more up-to-date data would allow researchers to gain a better understanding of the current situation and assess the effectiveness of new policies. Lastly, while the study provides valuable insights for policymakers, its focus is limited to China's specific context, and caution should be taken when applying the findings to other countries or regions with different social and economic characteristics. Therefore, future research could expand beyond China's context and test the generalizability of the current findings. Additionally, future studies could

employ more advanced econometric methods to address potential endogeneity issues and improve the robustness of the results.

**Author Contributions:** Conceptualization, M.C. and H.H.; methodology, M.C. and C.G.; software, M.C.; validation, Y.H., H.H. and M.C.; formal analysis, M.C.; investigation, H.H.; resources, H.H.; data curation, C.G.; writing—original draft preparation, M.C.; writing—review and editing, M.C. and B.A.; visualization, H.H.; supervision, H.H.; project administration, H.H.; funding acquisition, H.H. All authors have read and agreed to the published version of the manuscript.

**Funding:** This research was funded by National Social Science Fund General Project "Research on Integrated Development of Culture and Tourism in Ethnic Minority Areas to Promote Poverty Alleviation and Rural Revitalization" (21BKS026) and Topic of Social Science Achievement Review Committee of Hunan Province in 2022: Research on the Path of Green Revitalization of Rural Industries in Hunan Province under the Strategy of "Three High and Four New" (XSP22YBZ179).

**Acknowledgments:** We thank you very much the key project National Social Science Fund General Project and Topic of Social Science Achievement Review Committee of Hunan Province for the fund support to finalize our work.

**Conflicts of Interest:** The authors declare no conflict of interest.

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
