# Peer review of "Examining the Relationship between Transportation Infrastructure, Urbanization Level and Rural-Urban Income Gap in China"

_sustainability, doi:10.3390/su15108410_

Round 1

Reviewer 1 Report

See the pdf file

Nothing relevant

Author Response

Reviewer-1

Examining the relationship between transportation infrastructure,

urbanisation level and rural-urban income gap in China

Reviewer comments: This paper examines the relationship between the development of transport infrastructure and urban-rural income inequality in China. Based on its content, a possible alternative title could also be "Transportation Infrastructure and Urbanisation: Implications for Regional Economic Development and Income Disparities in China". Overall, the article finds that transport infrastructure has a significant negative effect on urban-rural income inequality, with the indirect effect greater than the direct effect, and that urbanisation plays an important mediating role in this relationship. This is not surprising, as it is in line with previous results in China and elsewhere. In fact, this is acknowledged in the literature review section, where it examines existing studies on transport infrastructure and income disparities. It also discusses how transport infrastructure can be a critical factor in economic development and how it can contribute to income inequality. The paper also highlights the importance of understanding the relationship between transport infrastructure and urbanisation to reduce income disparities.

Author response: Thank you for pointing the main findings of our paper

Reviewer comments: From a methodological perspective, the paper combines different approaches.

  1. A spatial-Dubin model used to measure and test the impact of transportinfrastructure on the urban-rural income gap. It takes into account the spatial dependence of the data and allows for the analysis of the direct and indirect effects of transport infrastructure on income inequality.

  1. An intermediate effect test method, to investigate the potential mediatingeffect of urbanisation in the relationship between transport infrastructure and income inequality.

  1. A panel data analysis from 30 provinces in China from 2010 to 2020 to analysethe impact of transport infrastructure on income inequality over time. The article also uses descriptive statistics to provide an overview of the data and to identify any trends or patterns in the data, and performs robustness checks to test the sensitivity of the results to different model specifications and to ensure the validity of the findings. Although the methods used could have been more sophisticated, I believe they are adequate to carry out the analysis. In fact, results indicate that transport infrastructure has a significantly negative direct, indirect, and total effect on the urban-rural income gap in China, with the indirect effect being larger than the direct effect. This suggests that transport infrastructure can effectively reduce income disparities with a significant spatial spillover effect. The level of urbanisation plays a significant mediating role in the effect of transport infrastructure on the urban-rural income gap, highlighting the role of transport infrastructure in improving urbanisation and reducing income disparities. These results are in line with the conclusions, focussing on the idea that the development of transport infrastructure can effectively reduce the urban-rural income gap in China. The study found that transport infrastructure has a significant negative effect on the urban-rural income gap, with the indirect effect being greater than the direct effect.

Author response: I appreciate your feedback on the methods and approach employed in our paper.

Reviewer comments: In summary, the main contributions of this paper are as follows.

  1. Investigates the impact of the development of transport infrastructure on theurban-rural income gap in China.

  1. Use a spatial Dubin model and an intermediary effect test method to analysethe impact of transport infrastructure on China's urban-rural income gap, and to examine the mediating role of the level of urbanisation.

  1. Covers a period from 2010 to 2020 and covers 30 provinces in China, usingpanel data for all 30 provinces, which allows a more comprehensive analysis of the development of the impact of transport infrastructure on the urban-rural income gap.

  1. Finds that transport infrastructure has a significant negative effect on theurban-rural income gap, with the indirect effect being larger than the direct effect, and that urbanisation plays an important mediating role in this 

  1. Highlights the importance of increasing both the level of urbanisation andcooperation between neighbouring regions to maximise the benefits of the development of the transport infrastructure to reduce income disparities and promoting regional balance in China.

Author response: Thank you for pointing the main contributions of our paper.

In my opinion, its main limitations include the following ones:

Reviewer comments: It focuses only on the relationship between transport infrastructure and income inequality and does not consider other factors that may also contribute to income inequality.

Author response: Thank you for highlighting this pertinent comment. As you mentioned, there are numerous factors that can both positively and negatively impact income inequality. However, our study was restricted by data limitations, so we concentrated solely on examining the possible influence of transportation infrastructure development and the mediating effect of urbanization.  

Reviewer comments: It only examines data from China and the findings may not be generalisable to other countries or regions.

Author response: Thank you for highlighting these relevant points. We also acknowledge that conducting cross-country investigations on the issue could yield valuable insights. Nevertheless, in our study, we focused solely on China's case, and we believe that countries exhibiting similar patterns in the factors included in this research may benefit from our findings.

Reviewer comments: It uses secondary data sources, which may have limitations in terms of accuracy and completeness.

Author response: Thank you for highlighting these essential comments. We also recognize that our study would have been more precise and comprehensive had we integrated primary data sources. However, the available secondary data sources provided sufficient data for us to conduct our study. We found that the data obtained from secondary sources allowed us to identify significant variables and comprehend the competitive landscape. Additionally, we recommend that future research endeavors incorporate primary data sources.

Reviewer comments: It does not consider the potential negative impacts of the development of the transport infrastructure, such as environmental degradation or displacement of local communities.

Author response: Thank you for highlighting this relevant comment. We also acknowledge that transportation infrastructure can have adverse impacts on the environment and local communities. However, our study solely investigated the relationship between transportation infrastructure and the rural-urban income gap. It can be inferred that future research studies that integrate and examine both positive and negative effects of transportation infrastructure on socio-economic development and environmental aspects can provide substantial findings to the existing literature.

Reviewer comments: It does not provide a detailed analysis of the mechanisms through which transport infrastructure affects income inequality, and more research is needed to explore these mechanisms in more detail. Possibly this is the main criticism. However, in general, I see no reason why the work should not be published.

Author response: Thank you for bringing this to our attention. Your exceptional comments and suggestions have encouraged us to broaden our research scope for future studies in this field.

Reviewer 2 Report

1.        The current literature review section is written more like a background introduction than a review of the research context. Perhaps this makes parts of the first two sections seem repetitive. Another point is that although this article puts forward the significance of the research, it does not clearly point out the gaps in the existing research.

2.        The last paragraph of Section 2.2 reviews that "There are also some potentially negative consequences to consider when investing in transportation infrastructure". Then, when Section 5 is given at the end, it would be better to combine the research on potentially negative consequences in other studies to give policy recommendations.

3.        It should be noted that the article seems to have some problems with some details.

a)       "gap" seems to be missed at the end of Title 2.1 "Transportation infrastructure development and its role in reducing rural urban income". And a hyphen was also missed between "rural" and "urban".

b)       Why is there a point in 2016 in Figure 1? No related description was found.

c)       Table 3 may be due to the fact that the table spreads across pages, causing errors in the horizontal divider.

Some sentences seem to have a deviation in emphasizing the content, such as lines 644 to 646 and 364 to 366.

In some paragraphs, such as lines 134 to 151, 206 to 215, and 676 to 686, there may be room for improvement in coherence and cohesion.

Author Response

Reviewer-2

Comments and Suggestions for Authors

Reviewer comment: The current literature review section is written more like a background introduction than a review of the research context. Perhaps this makes parts of the first two sections seem repetitive.

Author response: Thank you for highlighting this relevant comment. We have taken into account readers from other disciplines as well.

Reviewer comment:Another point is that although this article puts forward the significance of the research, it does not clearly point out the gaps in the existing research.

Author response: Thank you for pointing this out. We have inserted the gaps in existing studies as “The existing literature on the rural-urban income gap in China tends to overlook the relationship between transportation infrastructure, urbanization, and income disparities. This is a significant gap in the literature, as it fails to consider the interdependent nature of these factors and the potential implications of their integration. Addressing this gap by examining the interplay between transportation infrastructure, urbanization, and income disparities from multiple perspectives could provide a more comprehensive understanding of the issue and inform more effective policies and strategies for reducing rural-urban income disparities in China.

Reviewer comment: The last paragraph of Section 2.2 reviews that "There are also some potentially negative consequences to consider when investing in transportation infrastructure". Then, when Section 5 is given at the end, it would be better to combine the research on potentially negative consequences in other studies to give policy recommendations.

Author response: Thank you for this critical comment and we have added the following text in section 5 accordingly.

“While investments in transportation infrastructure can be beneficial, it is important to consider the potential negative consequences, including socioeconomic and environmental effects. To ensure that decision-makers have comprehensive information, future studies must also assess the risks associated with such investments. Therefore, it is imperative to take into account the negative impacts of transport infrastructure on society and the environment when making investment decisions.”

It should be noted that the article seems to have some problems with some details.

Reviewer comment: "gap" seems to be missed at the end of Title 2.1 "Transportation infrastructure development and its role in reducing rural urban income". And a hyphen was also missed between "rural" and "urban".

Author response: Thank you for bringing this to our attention. We have addressed the issues and made the necessary corrections.

Reviewer comment: Why is there a point in 2016 in Figure 1? No related description was found.

Author response: Thank you for pointing this out. It’s our mistake and removed it.

Reviewer comment: Table 3 may be due to the fact that the table spreads across pages, causing errors in the horizontal divider.

Author response: Thank you for bringing this to our attention. We have addressed the issues and made the necessary corrections.

Comments on the Quality of English Language

Reviewer comments: Some sentences seem to have a deviation in emphasizing the content, such as lines 644 to 646 and 364 to 366.

Author response: Thank you for pointing this out. We have replaced the sentences as follows.

  • Line 364 to 366

“Roads play a critical role in connecting rural and urban areas and are vital to overall economic development. They facilitate the movement of goods and people, which is essential to the production and sale of goods. In this study, highways were selected as the most representative type of traffic infrastructure for analysis.”

  • Line 644 to 646

“The results of the regression analysis, using various spatial weight matrices, reveal significant negative coefficients for transport infrastructure, with a statistical significance level of 1%. These findings suggest that investments in transport infrastructure can play a crucial role in reducing the income gap between urban and rural areas. This finding confirms the reliability and robustness of the spatial Dubin model applied in the study.”

Reviewer comments: In some paragraphs, such as lines 134 to 151, 206 to 215, and 676 to 686, there may be room for improvement in coherence and cohesion.

Author response: Thank you for bringing this to our attention. We have addressed the issues and made the necessary corrections.

  • Line 134 to 151

Numerous studies have investigated the factors that influence the income gap between rural and urban areas in China. One study discovered that as urbanization proceeded, both accumulated and flow income Gini coefficients declined, indicating a reduced rural-urban income gap (Ma et al., 2018). These findings suggest that governments could implement policies that encourage urbanization to narrow the income gap between rural and urban areas. Conversely, another study revealed that the direct effect of migration on household income was negatively associated with local resource contributions, highlighting the importance of developing local resources to boost rural residents' income (Yang et al., 2008). Entrepreneurial clusters, which are primarily composed of non-state-owned enterprises, are also capable of reducing local urban-rural income inequality by increasing rural residents' earnings. However, this clustering effect may be less pronounced in heavily urbanized regions or megacities (Guo et al., 2020). A recent study identified that the Provincial Per Capita Net Urban-Rural Income Ratio (PPUR) exhibited high spatial agglomeration in Eastern China but low values in Central and Western China. The PPUR in the province was influenced by factors such as industrial structure, infrastructure, medical resources, and land-centered urbanization. The rural-urban income gap boosted the province's PPUR but hindered it in nearby provinces, indicating a need for policies to narrow the gap and reduce the PPUR (Han et al., 2021).

  • Line 206 to 215

Empirical researchers have extensively studied the influence of transport infrastructure on the urban-rural income gap in China. Despite their effectiveness in reducing the gap, high-speed railways are not solely responsible for the formation of the three convergence clubs (Li et al., 2020). The impact of high-speed railways on narrowing the income disparity remains limited. Instead, national, provincial, and municipal roads play a significant role in reducing the urban-rural income gap in China (Lu et al., 2022). These roads facilitate rural labor mobility, providing access to local and regional job markets for migrant workers. Moreover, road infrastructure is particularly crucial for boosting the income of rural residents in China's southwestern and middle regions. With road access, these residents can participate in economic activities and increase their earnings. Consequently, roads are an essential factor in reducing the urban-rural income gap in these regions.

  • Line 676 to 686

Transportation infrastructure can also contribute to narrowing the urban-rural income gap by accelerating the pace of urbanization, consistent with research conducted by Mishra and Agarwal (2019). Financial support for agriculture, as a control variable, has a positive impact on the urban-rural income gap. However, it may face several challenges in the process, such as the allocation of multiple projects, decentralized distribution of funds, and limited management capacities, which could lead to insufficient funding for the agricultural sector. In contrast, foreign trade appears to have a significant impact on reducing the income gap between urban and rural areas. Firstly, foreign trade creates employment opportunities for rural residents, thereby increasing their wage income. Secondly, it provides farmers with a larger market to sell their agricultural products, increasing their productive income. Thus, foreign trade can help boost the income of rural residents and narrow the urban-rural income gap.”

Reviewer 3 Report

Dear Author, thanks for your work and I suggest the following notes to reconsider:

1. need to clarify the problem statement.

2. Add the average values of the Moran Index from 2010-2020 in Table 2.

3.  Clarify the Index range whether it is negative or positive.

4. Line 669 the author mention negative impact of variable for equ.3, which needs explanation, equ 3 is a positive correlation.

5. clarify equ. 3, symbols.

6. Add recommendations in separate paragraphs for the policymakers.

Author Response

Reviewer-3

Comments and Suggestions for Authors

Dear Author, thanks for your work and I suggest the following notes to reconsider:

Reviewer comments: need to clarify the problem statement.

Author response: Thank you for pointing this out. We have added the following text in the last part of introduction section. “Transportation infrastructure development is a crucial factor affecting regional economic growth and development. However, little is known about the relationship between transportation infrastructure development and income inequality in urban and rural areas of China.” The rest of the problem statement is described in this section.

Reviewer comments: Add the average values of the Moran Index from 2010-2020 in Table 2.

Author response:Thank you pointing this out. We have updated the table as follows.

Time

Moran's I index

P value

Average values of the Moran Index 

2010

0.402

0.000

0.306

2011

0.389

0.000

0.306

2012

0.386

0.000

0.306

2013

0.307

0.000

0.306

2014

0.295

0.000

0.306

2015

0.272

0.000

0.306

2016

0.258

0.000

0.306

2017

0.261

0.000

0.306

2018

0.258

0.000

0.306

2019

0.265

0.000

0.306

2020

0.275

0.000

0.306

2010

0.402

0.000

0.306

2011

0.389

0.000

0.306

2012

0.386

0.000

0.306

2013

0.307

0.000

0.306

2014

0.295

0.000

0.306

2015

0.272

0.000

0.306

2016

0.258

0.000

0.306

2017

0.261

0.000

0.306

2018

0.258

0.000

0.306

2019

0.265

0.000

0.306

2020

0.275

0.000

0.306

Reviewer comments: Clarify the Index range whether it is negative or positive.

Author response: Thank you for pointing this out. We have added the following text.

In table 2, the range of Moran's I indices is from 0.258 to 0.402, which suggests a moderate to strong positive spatial autocorrelation of the urban-rural income gap across provinces in China.” 

Reviewer comments: Line 669 the author mention negative impact of variable for equ.3, which needs explanation, equ 3 is a positive correlation.

Author response: Thank you for pointing this out. “We have explained in equ 3 as follows. In addition, if the regression coefficient in equation (3) γ1 Not significant γ2. Significant, indicating that the level of urbanization has a complete intermediary effect; If the regression coefficient γ1 and γ2 are significant, indicating that the level of urbanization has a partial intermediary effect.

Line 669…. “The results demonstrated statistical significance at a level of 1%, with regression coefficients showing a negative relationship (not significant).”

Reviewer comments: clarify equ. 3, symbols.

Author response: Thank you for the correction.

Road infrastructure (Roadit)

Urbanization (URit)

Other control variables (Xit)

Reviewer comments: Add recommendations in separate paragraphs for the policymakers.

Author response: Thank you for pointing this gap out. We have presented recommendations for policy makers in a separate paragraph ( see line 760-780) in the conclusion and policy recommendation section.
